# AD-GCN: A novel graph convolutional network integrating multi-omics data for enhanced Alzheimer's disease diagnosis

**Zilu Li**[1,2☯], **Qingyun Li**[3☯], **Xiaoqing Li**[2,4,5], **Wei Luo**[2,4,5,6], **Haiyan Guo**[2,4,5], **Chunyan Zhao**[3], **Canzhen Yang**[3], **Anke Xie**[7,8], **Kai Hu**[7,8,9*], **Yangfan Guo**[2,4,5*]

**1** Faculty of Information Engineering and Automation, Kunming University of Science and Technology, Kunming, China, **2** Central Laboratory, Yan'an Hospital Affiliated to Kunming Medical University, Kunming, China, **3** Neurology Department of Yan'an Hospital Affiliated to Kunming Medical University, Kunming, China, **4** Yunnan Key Laboratory of Tumor Immunological Prevention and Control, Kunming, China, **5** Precision Medicine Center, Yan'an Hospital Affiliated to Kunming Medical University, Kunming, China, **6** Kunming Medical University, Kunming, China, **7** Yunnan Innovation Institute of Beihang University, Kunming, China, **8** Yunnan Key Laboratory of Blockchain Application Technology, Kunming, China, **9** Institute of Computer Architecture (ICA), Beihang University, Beijing, China

☯ These authors contributed equally to this work.
* hukai@buaa.edu.cn (KH); guoyangfan@kmmu.edu.cn (YFG)

## Abstract

Alzheimer's disease (AD) etiology is complex, influenced by demographic risk factors such as age, sex, and educational level, alongside multi-omics factors derived from genomics, transcriptomics, and epigenomics. Advancements in multi-omics technology present both challenges and opportunities for AD diagnosis, enabling a more comprehensive understanding of the complex interactions among contributing factors, with the goal of improving diagnostic accuracy. To address this challenge, we propose a novel feature fusion approach in this study, AD-GCN, which integrates multi-omics data and their interaction networks to achieve more precise diagnosis and analysis of AD. In this study, we applied polygenic risk score and random forest algorithms for feature selection on genetic variation and methylation data. We then developed an AD-GCN for both multi-omics and single-omics classification tasks and compared its performance with that of machine learning ensemble methods. The experimental results demonstrated that multi-omics classification significantly outperformed single-omics classification, with AD-GCN surpassing the machine-learning ensembles. These findings highlight AD-GCN's strong potential to enhance AD diagnosis and improve accuracy in differentiating disease stages by integrating interactions across omics data, laying a solid foundation for the development of more precise and personalized AD diagnostic models.

**Data availability statement:** Raw data for this study were obtained from publicly available repositories: the Alzheimer's Disease Neuroimaging Initiative (ADNI, https://adni.loni.usc.edu/data-samples/adni-data/) and the Religious Orders Study and Memory and Aging Project (ROSMAP, available via Synapse: syn3219045, syn10901595, syn3157322, syn3157275 at https://www.synapse.org/Synapse). Access to ADNI and ROSMAP datasets requires free account registration. Genomic coordinates for methylation probes are available from the UCSC Genome Browser (https://genome.ucsc.edu/cgi-bin/hgSearch?db=hg19). All processed data, analysis code, and scripts used in this study are freely accessible via the GitHub repository: https://github.com/YananLAB/AD-GCN. In conclusion, all data needed to replicate our findings are freely and publicly available via established repositories (ADNI, ROSMAP, UCSC Genome Browser) and our GitHub repository.

**Funding:** This study was funded through the following: 1) National Key Research and Development Program (2022YFC2603305) (Awarded to YG); 2) Key R&D Project of Yunnan Province (202103AQ100002) (Awarded to YG); 3) Special Fund for the Central Government to Guide Local Science and Technology Development (202207AB110017) (Awarded YG); 4) Yunnan Academician Expert Workstation (202205AF150023) (Awarded to YG); 5) Yunnan Fundamental Research Projects (202201AY070001-192)(Awarded to YG); 6) Science and Technology Innovation Team of Kunming Medical University (CXTD202215) (Awarded to YG); 7) Yunnan Provincial Science and Technology Program (202302AF080006) (Awarded to KH).

**Competing interests:** The authors have declared that no competing interests exist.

## Introduction

Alzheimer's disease (AD) is a progressive neurodegenerative disorder and the leading cause of dementia worldwide. In China, the aging population has led to a significant rise in AD cases, with over 9 million individuals affected as of 2020 [1]. This growing prevalence underscores the urgent need for improved diagnostic and therapeutic approaches. AD involves a variety of risk factors, which can be divided into two main categories: non-modifiable risk factors, including age, sex, genetic factors, and family history, and modifiable risk factors, such as history of cardiovascular diseases, unhealthy diet, sleep disturbances, and education level [2]. Large-scale genome-wide association studies (GWAS) have identified AD genetic risk factors such as *APOE* [3]. Furthermore, DNA methylation is an epigenetic modification involving the selective addition of a methyl group to cytosine of the CG dinucleotide in DNA sequences, which can change genetic expression without altering the DNA sequence itself. Altered DNA methylation patterns in specific genes involved in AD-related processes, such as inflammation and protein degradation, can disrupt neuronal function and contribute to increased AD risk [4].

Research on AD and its early stage, mild cognitive impairment (MCI), increasingly utilizes diverse data sources to gain a deeper understanding of the disease [5–7]. Clinical data and cognitive assessments provide demographic information and symptom phenotype data of patients. Genomics, through the analysis of genetic variations, identifies individuals at increased risk for AD by examining individual polymorphisms. Epigenomics, particularly DNA methylation profiling, reveals alterations in gene regulation that may contribute to AD pathogenesis. Collectively, these multi-omics data illuminate the complex biological mechanisms underlying AD and MCI from different aspects.

The fusion of multi-omics data is gradually becoming a cutting-edge approach for the precise diagnosis of AD [8–12]. Early multi-omics integration methods were primarily based on traditional machine learning (ML) models, encompassing three principal strategies: feature concatenation-based, transformation-based, and model-based integration. Specifically, concatenation-based integration directly combines multi-modal feature vectors as classifier inputs; transformation-based integration performs dimensionality reduction/feature transformation prior to fusion; model-based integration independently trains single-omics models before aggregating outputs. However, these conventional methods fail to account for inter-omics correlations, potentially introducing modality-specific biases. Recent advances have witnessed the emergence of graph convolutional networks (GCN) as powerful deep learning tools for biomedical multi-omics classification, demonstrating superior capability in modeling complex network-structured data relationships [13,14]. Schulte-Sasse et al. [15] developed an interpretable GCN framework integrating pan-cancer multi-omics data (mutations, copy number variations, DNA methylation, and gene expression) with protein-protein interaction networks for cancer gene prediction. Wang et al. [16] leveraged GCN and sample similarity metrics to exploit cross-omics label-space correlations for tumor classification. These approaches effectively leverage the graph-structured properties of GCN, which (1) represent inter-omics topological

relationships as graph networks and (2) elucidate cross-omics feature dependencies through adaptive learning of edge weights. Consequently, the development of GCN-based diagnostic models capable of comprehensively integrating multi-omics data while exploiting inter-omics interactions presents a highly promising research direction.

This study proposes a novel multi-omics data integration method, AD-GCN, which integrates multi-omics data to enhance the accuracy of AD diagnosis and the ability to differentiate disease stages. As illustrated in **Fig 1**, we gathered clinical, genetic variation, and methylation omics data, which were preprocessed for cleansing. Feature selection or dimensionality reduction has been applied across different omics data types. Different ML ensemble strategies were then used on both single- and multi-omics datasets to validate the classification performance of the selected features. Ultimately, the AD-GCN model was developed and compared with traditional ML models to assess its efficacy in integrating the three types of omics data for diagnostics.

The specific *aims* of this paper are as follows

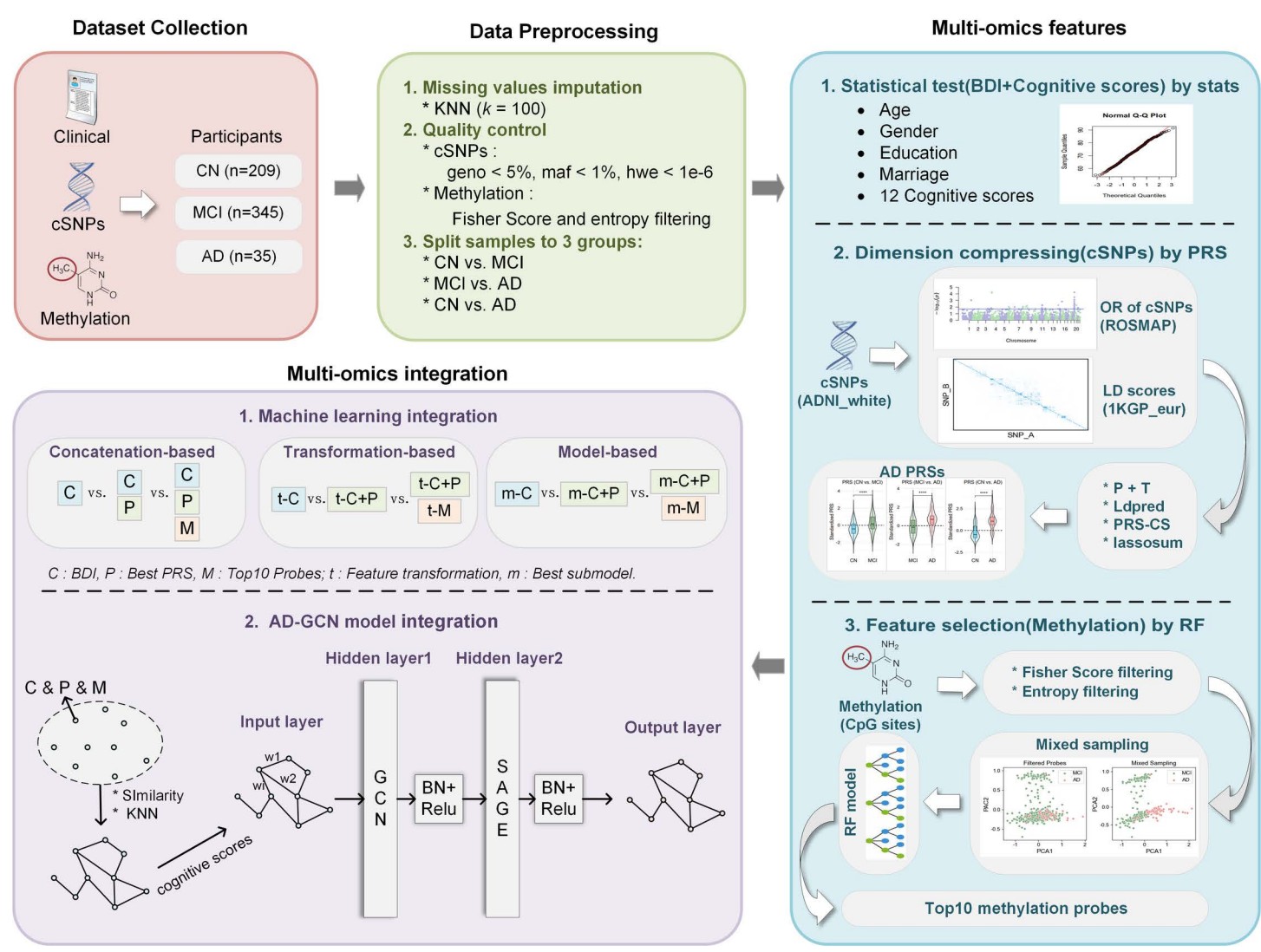

**Fig 1. Workflow diagram of AD precise diagnosis in this study.**

- We enhanced AD prediction by integrating multi-omics data—including clinical assessments, genetic variation, and methylation profiles—and applying effective Polygenic Risk Scores (PRS) and feature selection-based dimensionality reduction for data.

- To tackle class imbalance in methylation data, we combined borderline-SMOTE with borderline undersampling to enhance boundary clarity by removing majority instances and augmenting minority instances near the boundary.

- We developed the AD-GCN, which constructs edges based on sample similarity and k-nearest neighbors, and use cognitive score correlations as edge weights to form the adjacency matrix, outperforming ML ensemble methods.

## Materials and methods

### 1. Data acquisition

This study employed a heterogeneous dataset, comprising 589 Caucasian individual samples, for multi-omics feature fusion analysis. The Alzheimers Disease Neuroimaging Initiative (ADNI) dataset (https://adni.loni.usc.edu/) [17] encompasses three types of data: clinical data, genomic variants (SNPs), and DNA methylation profiles. Clinical data included basic demographic information (BDI) such as age, sex, educational level, and marital status, along with 12 cognitive function test scores (e.g., CDRSB, ADAS11, MMSE, RAVLT, etc.). Detailed descriptions of these cognitive tests are provided in S1 Table in S1 File. Genomic variants were obtained by whole-genome sequencing (WGS). DNA methylation profiles were obtained using an Illumina Infinium Human Methylation EPIC Bead Chip (850k).

The Religious Orders Study and Memory and Aging Project (ROSMAP) dataset [18] was obtained from the Synapse (https://www.synapse.org/Synapse:syn3219045) with appropriate data access permissions. This dataset was used in GWAS analyses, providing effect sizes for each SNP site under different disease stages for the samples in the ADNI cohort. All the datasets were primarily derived from Caucasian individuals. A search in the GWAS Catalog using the keyword "Alzheimer's disease" identified 2,936 genetic loci associated with AD risk. After overlapping these loci with data from ADNI, 2,242 candidate SNPs (cSNPs) were retained for further investigation.

The 1000 Genomes Project (1KGP) [19] sequenced over 2,500 individuals from 26 global populations, providing a diverse Linkage disequilibrium (LD) reference for genetic studies. In our study, we focused on the European subset of 1KGP to match the ADNI cohort, ensuring population-specific polygenic risk score (PRS) calculations and reducing stratification bias.

### 2. Data preprocessing

Based on clinical diagnoses within the ADNI and ROSMAP datasets, this study encompassed three patient groups: CN (Cognitively Normal), MCI, and AD, and analyzed them in three pairwise comparisons: CN vs. MCI, MCI vs. AD, and CN vs. AD. The distribution of cases in the ADNI and ROSMAP cohorts is summarized in **Table 1**. Missing values in the clinical data were filled out using the K-nearest neighbor (KNN) algorithm, and statistical differences between each group were analyzed using the Kruskal-Wallis test and chi-square test, implemented in the R software (version 4.3).

All 2242 cSNPs were extracted from the ADNI and ROSMAP cohorts, retaining only those that overlapped between the two datasets. SNP quality control was performed using the Plink tool, excluding SNPs with (1) a low detection rate (< 5%), (2) a low minor allele frequency (< 1%), and (3) a significant deviation from the Hardy-Weinberg equilibrium ($p < 1e$-6).

**Table 1. Case Distribution in ADNI and ROSMAP Cohorts.**

| Cohort | CN (Cognitively Normal) | MCI (Mild Cognitive Impairment) | AD (Alzheimer's Disease) | Total |
|--------|-------------------------|----------------------------------|--------------------------|-------|
| ADNI | 209 | 345 | 35 | 589 |
| ROSMAP | 379 | 279 | 495 | 1,153 |

The detection rate threshold of < 5% was selected to exclude SNPs with poor genotyping quality, which may result from technical issues such as low hybridization efficiency or insufficient signal intensity. This threshold ensures data completeness and minimizes the impact of missing data on subsequent analyses [20]. The minor allele frequency (MAF) threshold of < 1% was chosen to retain a broader range of genetic variants while minimizing the inclusion of rare variants that may result from sequencing errors or technical noise [21,22]. The Hardy-Weinberg equilibrium (HWE) threshold of $p < 1e-6$ was applied to exclude SNPs with significant deviations, which may indicate genotyping errors, population stratification, or natural selection [23]. These thresholds are widely adopted in genetic studies to ensure data quality and reliability.

### 3. Risk scoring of cSNPs

A meta-analysis of GWAS summary statistics from the ROSMAP genomics dataset was conducted to identify SNPs associated with different AD disease states using three pairwise comparisons. To account for potential confounding effects of demographic factors, age, sex, and education level were included as covariates in the analysis. Principal Component Analysis (PCA) was applied to the genotype data from cSNPs to identify the population structure, and the top three genotype principal components were used as covariates to control population stratification.

Four common PRS methods were applied to assess genetic risk across different comparison groups [24,25]: Pruning and Thresholding (P + T) [26,27], LDpred [28], PRS-CS [29], and lassosum [30]. LD scores calculated from the 1KGP [31] European population, serving as the LD panel for the ADNI Caucasian population [32]. This LD panel was essential for methods such as LDpred, PRS-CS, and lassosum, which rely on LD information to account for the correlation structure between genetic variants and optimize the accuracy of PRS calculations. To evaluate the predictive ability of the different PRS methods, the calculated PRSs were combined with the BDI and used as features in a logistic regression (LR) model. Model performance was assessed using Nagelkerke's $R^2$ and area under the receiver operating characteristic curve (AUC).

To assess the relationship between the best-performing PRS and cognitive function, we evaluated the association between the PRS and 12 clinical cognitive function scores across different disease groups. The Kolmogorov-Smirnov (KS) test was used to compare the distributions of cognitive scores between groups defined by the PRS. Statistically significant differences ($p < 0.05$) indicated a robust association between the PRS and cognitive performance.

### 4. Methylation feature selection

The methylation 850k data underwent quality control following the standard procedures of the Chip Analysis Methylation Pipeline (ChAMP) [33]. The specific steps included: (1) extracting raw beta values from the original methylation signals; (2) filtering out probes of poor quality; (3) imputing missing values using KNN ($k = 100$); (4) adjusting for distribution differences between Type I and Type II probes using the BIMQ method [34]; (5) correcting $P$-values using the Benjamini-Hochberg algorithm [35], selecting probes with a corrected $P$-value of less than 0.001; and (6) selecting probes located on CpG islands.

To address class imbalance, a hybrid sampling algorithm integrating borderline-SMOTE, a random oversampling method that increases minority class samples at the decision boundary [36], and borderline-undersampling were applied to balance the data. Subsequently, we employed filter and embedded methods for feature selection on the balanced dataset. In the filter approach, Fisher Score and entropy-based filtering methods were utilized to identify features with minimal intra-class variance and maximal inter-class variance, thereby effectively eliminating less informative probes. For the embedded method, we applied Random Forest (RF) to rank the importance of remaining probes in each comparison group, selecting the top 50 most significant probes as input features for the Support Vector Machine (SVM) model. Model performance was subsequently evaluated using G-mean, accuracy, precision, and F1-score metrics.

### 5. Multi-omics feature fusion method

**5.1. Machine learning techniques.** Three machine learning integration strategies—concatenation-based integration, transformation-based integration, and model-based integration—were implemented and evaluated to

identify the most effective approach for multi-omics data [37]. In concatenation-based integration, clinical and genetic variants and methylation features are directly combined and used as inputs for eight common ML classification models. Transformation-based integration involved applying PCA to reduce the dimensionality of the BDI, BDI+PRS, and top 10 methylation probes data for each comparison group, with reduced dimensions set to 3, 4 and 5, respectively. Finally, model-based integration uses separate classifiers for each omics data with five widely-used ML methods (KNN, LR, Bayes, SVM and RF), selected for their diverse learning paradigms and fundamental nature in machine learning. Optimal classifiers were chosen based on negative log-loss scores and combined using soft voting for the final prediction. The performance of each integration strategy was evaluated using the Matthews Correlation Coefficient (MCC), accuracy, and precision.

To ascertain the incremental value of multi-omics integration and evaluate the individual and collective contributions of each data modality to diagnostic accuracy, a systematic comparison of the classification performance of models was performed using: (1) clinical data (BDI) exclusively; (2) clinical data in conjunction with the optimal PRS (genomics); and (3) clinical data, PRS, and the top 10 methylation probes (epigenomics).

**5.2.  Novel graph convolutional neural network.**  The AD-GCN model consisted of four layers: an input layer, two hidden layers, and an output layer. In the input layer, an adjacency matrix is constructed using the following two steps. First, a composite similarity matrix was calculated from concatenated multi-omics features, and sample pairs with a similarity greater than 0.5 were retained. Second, KNN was applied with optimal hyperparameters to expand the range of associated sample pairs and further refine the network structure. The degree of association between sample pairs was measured using the similarity of 12 cognitive scores, with weighted edges representing the strength of these relationships. The two hidden layers utilize a graph convolutional kernel and a GraphSAGE kernel, reducing the input data to eight and three respectively. To enhance network stability, the output of the hidden layers was processed through a ReLU activation function, gradient clipping, and batch normalization.

Ablation experiments were conducted based on the ADNI dataset, incorporating the following three strategies. (1) Strategy1: Using the sample similarity matrix calculated from multi-omics features as the adjacency matrix, while employing the cognitive score correlation matrix as edge weights. (2) Strategy2: Constructing the adjacency matrix using both the sample similarity matrix and K-nearest neighbors, with all edge weights set to 1. (3) Strategy3: Constructing the adjacency matrix using both the sample similarity matrix and K-nearest neighbors, while employing the cognitive score correlation matrix as edge weights.

## Results

### 1.  Cognitive scores as clinical prior knowledge

We evaluated clinical differences in the ADNI dataset within the three groups (CN vs. MCI, MCI vs. AD, and CN vs. AD) using the BDI and 12 cognitive function scores. As summarized in **Table 2**, the CN group was slightly older on average than the MCI and AD groups ($p = 0.002$), and the proportion of females in the MCI and AD groups was notably higher than that in the CN group ($p = 0.045$), while no statistically significant differences were observed in years of education and marital status across the three groups. Additionally, we observed statistically significant differences ($p < 0.005$) in all 12 cognitive function scores across disease states. These findings suggest that BDI (including age, sex, education level, and marital status) alone are not sufficient to differentiate between disease states, whereas cognitive function emerged as a key differentiator.

While 12 cognitive scores were analyzed and found to be informative, it is rare to collect all of them in routine clinical practice. Therefore, considering clinical applicability and model simplicity, the BDI was used as the primary clinical data, with insights from the 12 cognitive scores serving as prior knowledge to support the development of subsequent analytical models.

**Table 2. Summary of the ADNI participants.**

| | CN | MCI | AD | *P*-value |
|---|---|---|---|---|
| **Age in years** | 74.61(5.54) | 72.4(7.39) | 74.14(8.91) | 0.0021[a] |
| **Years of education** | 16.0(14.0-18.0) | 16.0(14.0-18.0) | 16.0(14.0-18.0) | 0.4420[a] |
| **Gender** | | | | 0.0445[b] |
| Male | 102(48.8%) | 205(59.4%) | 21(60%) | |
| Female | 107(51.2%) | 140(40.6%) | 14(40%) | |
| **Status of marriage** | | | | 0.2343[b] |
| Never married | 7(3.3%) | 6(1.7%) | 0(0%) | |
| Married | 151(72.2%) | 271(78.6%) | 31(88.6%) | |
| Divorced | 20(9.6%) | 33(9.6%) | 1(2.9%) | |
| Widowed | 31(14.8%) | 35(10.1%) | 3(8.6%) | |
| **CDRSB** | 0.0(0.0-0.0) | 1.5(1.0-2.0) | 4.5(3.75-5.5) | <0.0001[a] |
| **ADAS11** | 5.33(3.67-7.0) | 9.0(6.0-12.0) | 18.0(14.5-22.5) | <0.0001[a] |
| **ADAS13** | 9.0(6.0-11.0) | 14.67(10.0-19.67) | 28.0(24.0-32.5) | <0.0001[a] |
| **ADASQ4** | 3.0(2.0-4.0) | 5.0(3.0-7.0) | 9.0(8.0-10.0) | <0.0001[a] |
| **MMSE** | 29.0(28.0-30.0) | 28.0(27.0-29.0) | 23.0(21.0-24.5) | <0.0001[a] |
| **RAVLTimmediate** | 45.0(38.0-52.0) | 35.0(28.0-44.0) | 22.4(17.0-26.5) | <0.0001[a] |
| **RAVLTlearning** | 6.0(4.0-8.0) | 4.0(3.0-6.0) | 2.0(1.0-3.5) | <0.0001[a] |
| **RAVLTforgetting** | 3.0(2.0-5.0) | 4.0(3.0-6.0) | 5.0(3.0-5.5) | 0.0038[a] |
| **RAVLTpercforgetting** | 33.3(16.7-50.0) | 50.0(30.0-83.3) | 100.0(80.6-100.0) | <0.0001[a] |
| **LDELTOTAL** | 13.0(11.0-15.0) | 7.0(4.0-9.0) | 1.0(0.0-4.0) | <0.0001[a] |
| **TRABSCOR** | 75.0(56.0-93.0) | 87.0(66.0-121.0) | 228.0(146.08-283.5) | <0.0001[a] |
| **FAQ** | 0.0(0.0-0.0) | 1.0(0.0-4.0) | 15.0(10.0-18.5) | <0.0001[a] |

[a]Kruskal-Wallis test,

[b]Chi-squared test. *Abbreviations: CDRSB* Clinical Dementia Rating (Sum of Boxes), *ADASQ4* Alzheimer's Disease Assessment Scale (Question 4), *ADAS11* Alzheimer's Disease Assessment Scale (11-item Cognitive Subscale), *ADAS13* Alzheimer's Disease Assessment Scale (13-item Cognitive Subscale), *MMSE* Mini-Mental State Examination, *RAVLTimmediate* Rey Auditory Verbal Learning Test (Immediate Recall), *RAVLTlearning* Rey Auditory Verbal Learning Test (Learning), *RAVLTforgetting* Rey Auditory Verbal Learning Test (Forgetting), *RAVLTpercforgetting* Rey Auditory Verbal Learning Test (Percentage Forgetting), *LDELTOTAL* Logical Memory Delayed Recall (Total Score), *TRABSCOR* Trail Making Test Part B (Total Score), *FAQ* Functional Activities Questionnaire.

## 2. LDpred PRS effectively distinguishes disease and cognitive states

**2.1. Maximum explanatory power in LDpred PRS.** To ensure comparability, standardized quality control was performed on genotype data from the ADNI, ROSMAP, and 1KGP cohorts. For sample quality control, we focused on Caucasian participants from three cohorts to ensure a consistent ethnic background. PCA confirmed this homogeneity, demonstrating close alignment of genetic variation in the ADNI and ROSMAP cohorts with the European population within 1KGP (**Fig 2**). Loci with low detection rates, low minor allele frequencies, or significant deviations from Hardy-Weinberg equilibrium were removed to ensure data quality and to identify potential genotyping errors or selection bias. Following quality control, 1,322 overlapping SNPs were identified from 2,242 SNPs across the three merged cohorts.

To calculate the PRSs corresponding to the three ADNI comparison groups, ROSMAP samples were divided into the same three comparison groups as in the ADNI dataset: CN vs. MCI (n = 676), MCI vs. AD (n = 792), and CN vs. AD (n = 874). We used four different models of PRS: P + T, LDpred, PRS-CS, and lassosum to assess their ability to differentiate between CN, MCI, and AD. Model performance was assessed using Nagelkerke's $R^2$, a modification of the Cox and Snell $R^2$ which provides a measure of the model's explanatory power ranging from 0 to 1 for categorical variables. Higher $R^2$ values indicate better fit and greater explanatory power.

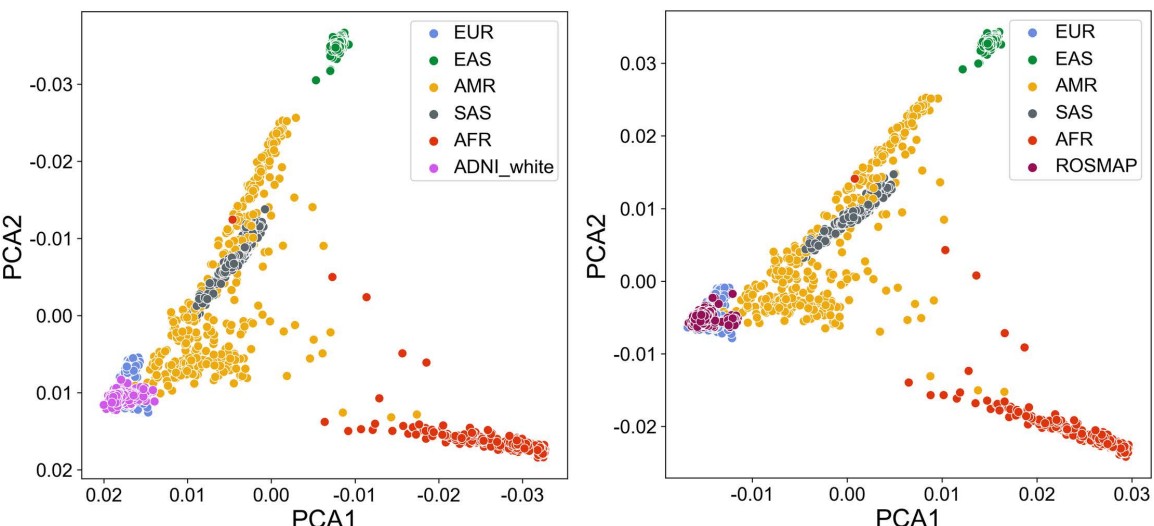

**Fig 2. PCA distribution of Caucasian populations from ADNI and ROSMAP cohorts in the 1KGP dataset.**

The P + T model achieved its highest predictive power using 29 (CN vs. MCI), 72 (MCI vs. AD), and 65 (CN vs. AD) SNPs with partition thresholds set at $p_T < 0.0191$, $p_T < 0.0468$, and $p_T < 0.0306$, respectively. Nagelkerke's $R^2$ values were 0.015, 0.058, and 0.031, respectively, indicating variance explanatory capabilities between 1% and 6%. The LDpred, PRS-CS, and lassosum models required LD scores from the 1KGP. A strong correlation ($r = 0.983$) between the LD calculations for the European population of 1KGP and the Caucasian population of ADNI validated the use of 1KGP as an external LD reference panel (S1 Fig **in** S1 File). For these models, Nagelkerke's $R^2$ values varied across three diagnostic comparisons (CN vs. MCI, MCI vs. AD, and CN vs. AD). LDpred achieved the highest $R^2$ values (0.09–0.313), followed by lassosum (0.016–0.186) and PRS-CS (0.004–0.050) (Figs 3–6).

Across all four models, the CN vs. AD comparison generally yielded the highest $R^2$ values, followed by MCI vs. AD and CN vs. MCI. These results suggest that the performance of PRS in predicting AD is model-dependent and varies across disease stages. LDpred demonstrated significantly greater explanatory power for AD risk, particularly in differentiating between cognitively normal individuals and those with AD.

**2.2. Optimal diagnostic performance with LDpred PRS.** To evaluate the diagnostic utility of PRS, we performed a series of classification experiments using an LR model. Although PRS can be used independently for classification, its predictive capacity is limited, as it captures only genetic risk information. Therefore, we integrated the PRSs with the BDI to assess whether combining genetic and biomarker information could improve diagnostic accuracy. Specifically, we investigated the impact of different PRS models (P + T, LDpred, PRS-CS, and lassosum) on the classification performance of BDI across three diagnostic comparisons: CN vs. MCI, MCI vs. AD, and CN vs. AD. Incorporating PRS generally improved the diagnostic classification compared with using BDI alone, as reflected by the increased AUC values. As illustrated in Table 3, when the BDI was combined with the four different PRS models in the CN vs. MCI group, the AUC values increased by 1% (P + T PRS), 0% (PRS-CS PRS), 1% (lassosum PRS) and 6% (LDpred PRS). In the MCI vs. AD group, AUC improvements of 7% (P + T PRS), 8% (PRS-CS PRS), 5% (lassosum PRS) and 15% (LDpred PRS) with the integration of PRSs. Similarly, in the CN vs. AD comparison, combining BDI with the four PRS models resulted in AUC enhancements of 10% (P + T PRS), 2% (PRS-CS PRS), 20% (lassosum PRS) and 27% (LDpred PRS). The greatest improvement was observed in the CN vs. AD group (average increase 14.6%), whereas the CN vs. MCI group showed the smallest improvement (2%).

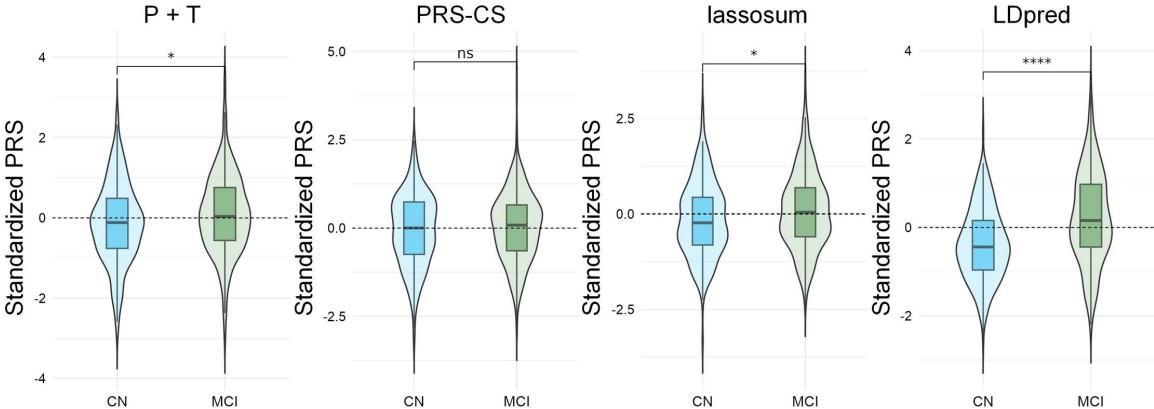

**Fig 3. Kernel density plots of four PRS models in CN vs. MCI group.**

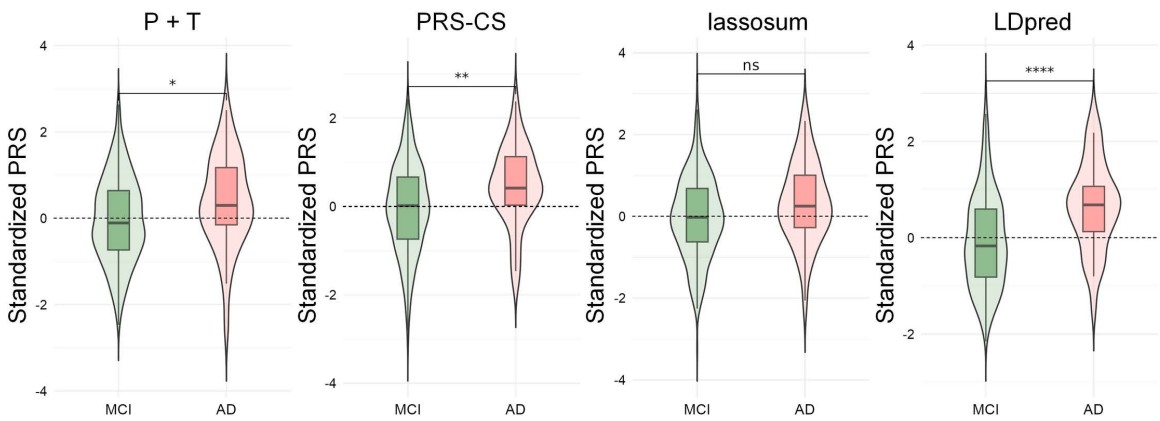

**Fig 4. Kernel density plots of four PRS models in MCI vs. AD group.**

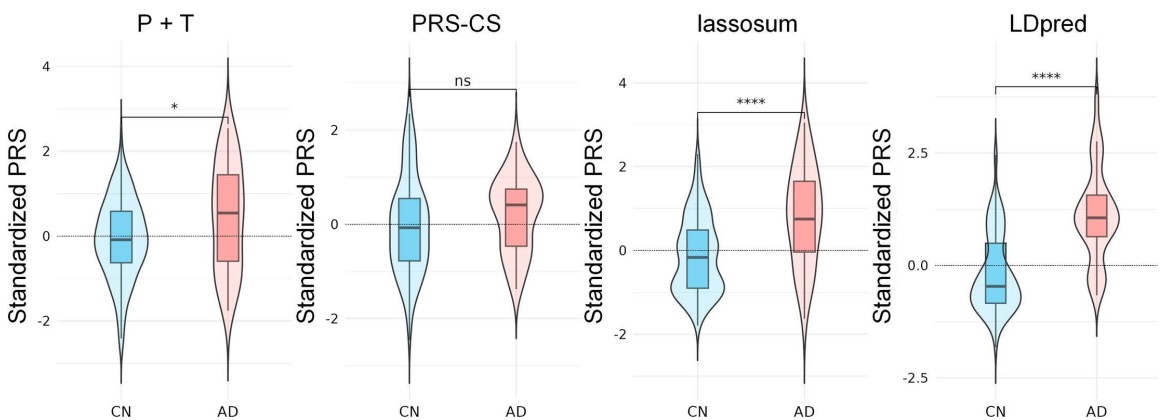

**Fig 5. Kernel density plots of four PRS models in CN vs. AD group.**

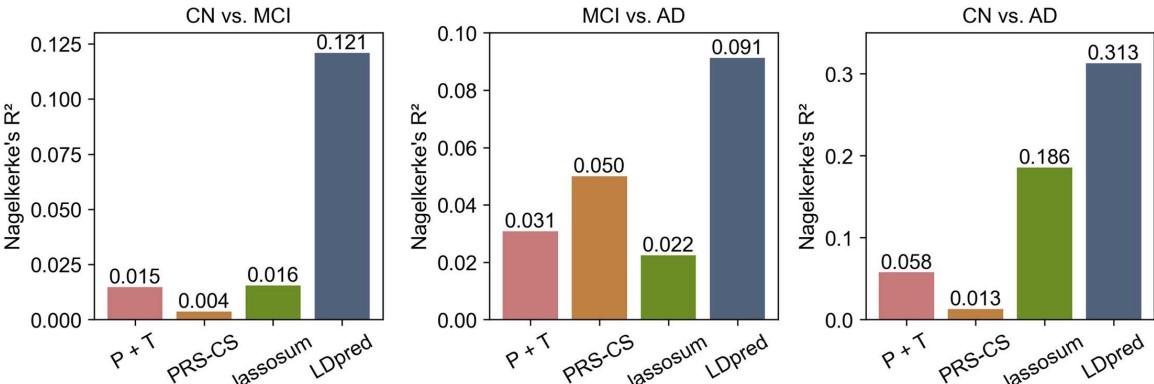

**Fig 6. Comparison of variance explained by four PRS models.**

**Table 3. Average AUC value under five-fold cross-validation for the four PRS models.**

| Group | BDI | BDI_P+T | BDI_PRS-CS | BDI_lassosum | BDI_LDpred |
|---|---|---|---|---|---|
| CN vs. MCI | 0.592 | 0.598 | 0.593 | 0.604 | 0.651 |
| MCI vs. AD | 0.544 | 0.605 | 0.625 | 0.588 | 0.685 |
| CN vs. AD | 0.519 | 0.616 | 0.539 | 0.717 | 0.787 |

Across all PRS models, both the explained variance (Nagelkerke's $R^2$) and classification performance (AUC in the LR model) showed a consistent trend: lowest for CN vs. MCI, intermediate for MCI vs. AD, and highest for CN vs. AD. This excellent ability of PRS in distinguishing CN from AD stems from the polygenic genetic nature of AD. As a highly inherited disease, AD is driven by both high-effect variants such as *APOE-ε4* and numerous small-effect loci. PRS integrates these genetic effects to give AD patients a significantly higher cumulative risk than the CN population with a protective or neutral genetic background.

The LDpred model demonstrates superior predictive performance, exhibiting significantly higher values for both Nagelkerke's $R^2$ and average AUC metrics. From the perspective of shrinkage mechanisms, P+T retains significant SNPs through hard thresholding but loses weak-effect signals, while PRS-CS employs global continuous shrinkage that excessively compresses moderate-effect variants. Lasso-type methods like lassosum analyze only partial SNPs in high LD regions through regularization. In contrast, LDpred achieves localized adaptive shrinkage by explicitly modeling LD structures, which not only coordinates the adjustment of SNP effect sizes within high-LD regions but also preserves subthreshold genetic signals, thereby demonstrating higher variance explained and classification accuracy in AD risk prediction. The superior performance of the LDpred model justified its selection as a representative genomic feature for subsequent analysis.

**2.3. Distinct cognitive disparities revealed by LDpred PRS.** Linear fits based on LDpred PRS for 12 cognitive scores across different disease stages showed that the KS values for each comparison group were greater than zero (all $P<0.001$ except RAVLTpercforgetting), indicating distinct distributions of standardized PRS between groups and statistically significant differences in genetic risk across disease stages (S2-S4 Figs in S1 File). Moreover, the minimum standardized PRS values in patients with more severe disease progression were consistently higher than those in patients with milder disease stages.

Using ADAS11 as a representative cognitive measure, KS statistic showed a stepwise increase with disease progression: 0.41 (CN vs. MCI), 0.72 (MCI vs. AD), and 0.97 (CN vs. AD) (all $P<0.001$). This monotonically increasing pattern of KS values suggests that greater genetic risk load is associated with greater cognitive decline. It is worth noting that the

minimum value of standardized PRS corresponding to the leftmost end of the fitting curve (min-sPRS) shows a consistent directional change rule: min-sPRS (MCI)> min-sPRS (CN) in the comparison of CN vs. MCI; min-sPRS (AD)> min-sPRS (MCI) in MCI vs. AD; min-sPRS (AD)> min-sPRS (CN) in CN vs. AD. These findings indicate that the genetic risk gradient represented by LDpred PRS increases with the degree of clinical cognitive decline, and the statistical significance ($P<0.001$) confirms the strong discriminative power of all group comparisons.

### 3. Effective extraction of methylation markers on balanced datasets

To investigate the role of epigenetic risk factors in AD progression, we analyzed the methylation data corresponding to three comparison groups: CN vs. MCI, MCI vs. AD, and CN vs. AD. Raw methylation data were processed using the ChAMP package in R, the specific parameters are detailed in the Data preprocessing section. Because methylation changes within CpG islands are strongly associated with gene expression and are critical for differential methylation analysis, we restricted our analysis to probes located within the CpG islands. This focused feature selection yielded 2,512 probes (CN vs. MCI), 961 probes (MCI vs. AD), and 942 probes (CN vs. AD).

We applied the Fisher Score and entropy filtering methods to identify the most informative methylation probes. We plotted the distribution of Fisher Scores (S5–S7 Figs in S1 File), which approximates a normal distribution. Therefore, we selected the top 25% as the threshold, a commonly used and moderate strategy that removes low-scoring features while retaining probes with higher information content. In the Fisher Score-filtered probe set, we further applied entropy-based filtering to remove probes with zero information gain. A comparison showed that, across the three groups, the number of probes with information gain greater than zero was similar to the number of probes within the top 25%–100% quartile range (S5 Table in S1 File). Given the minimal differences between the two methods, we adopted the 25% threshold for both to streamline the process and maintain consistency. The number of methylation features retained in each group were 1,161 (CN vs. MCI), 422 (MCI vs. AD), and 472 (CN vs. AD) (S6 Table in S1 File). The remaining features are considered crucial for distinguishing between the different disease states.

The Imbalance Ratio (IR) of these key methylation probe data ranges from 1.69 to 9.86 across comparison groups, indicating a significant class imbalance issue. This imbalance could bias the feature selection towards the majority class. A hybrid sampling method combining borderline-SMOTE and borderline-undersampling was used to balance the data, which effectively reduced IR to 1.03–1.65. PCA demonstrated a clearer decision boundary between classes after balancing (Fig 7), demonstrating the effectiveness of this approach in both reducing IR and removing potentially noisy or ambiguous data points at the class boundaries.

We used an RF model with default parameters to rank the methylation features from a balanced dataset based on their importance. The top 50 features that contributed to the model were selected and evaluated using an SVM classifier. Cross-validation revealed that all top 50 features contributed to the classification performance (Fig 8). However, using only the top 10 features yielded a performance comparable to that obtained using all 50 features (average difference in evaluation scores of approximately 0.1). Therefore, for simplicity and computational efficiency, we selected the top 10 probes as representative epigenetic features for each comparison group (Fig 9 and S7 Table in S1 File).

### 4. AD-GCN outperforms ML in multimodal data integration

Given the complex interplay of genetic, epigenetic, and clinical factors in AD, we employed a multi-omics integration approach to improve the predictive accuracy and gain deeper insights into disease mechanisms. We conducted classification experiments using three machine learning-based ensemble strategies: concatenation-based, transformation-based, and model-based approaches. To evaluate the added benefits of multi-omics integration for AD prediction, we compared the performance of three increasingly complex strategies using an ensemble ML approach: (1) a clinical model using only BDI (C); (2) a model combining clinical and genetic data (BDI and LDpred PRS; C+P); and (3) a multi-omics model integrating clinical, genetic, and epigenetic data (BDI, LDpred PRS, and the top 10 methylation sites; C+P+M).

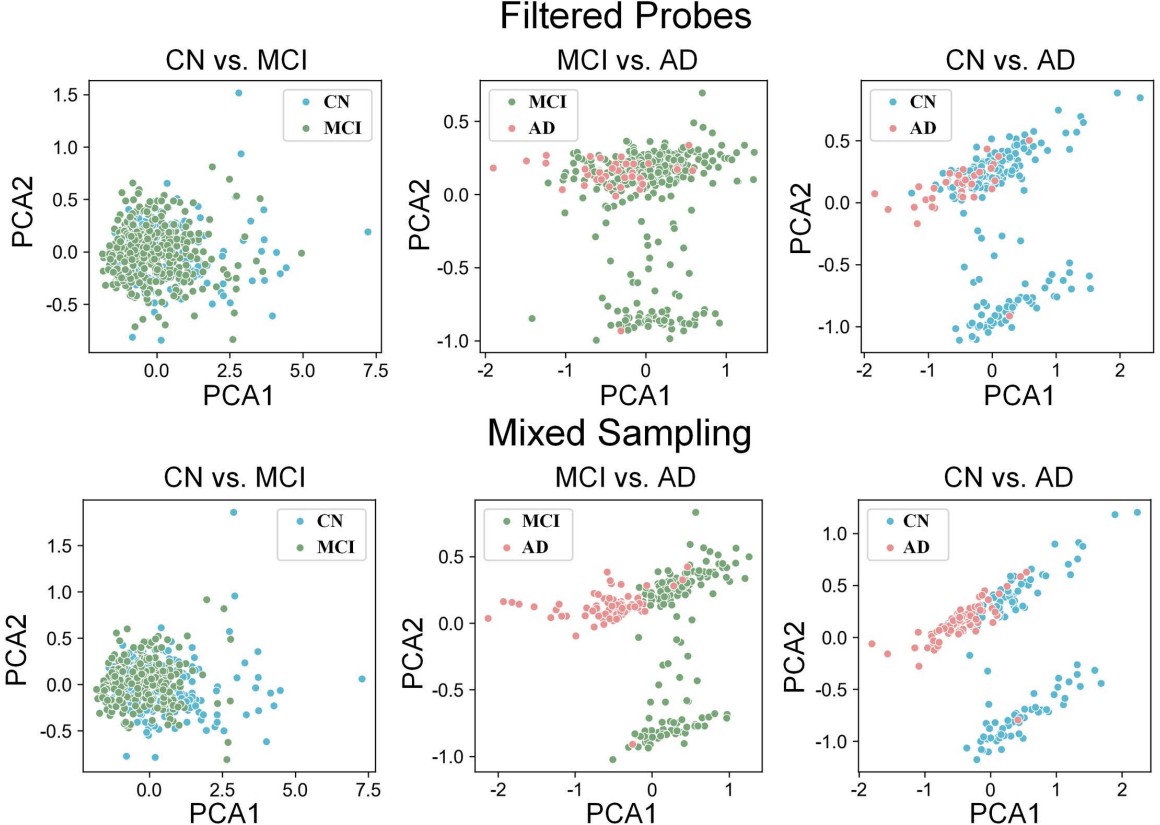

**Fig 7. PCA distribution of three comparative groups before and after data balancing.**

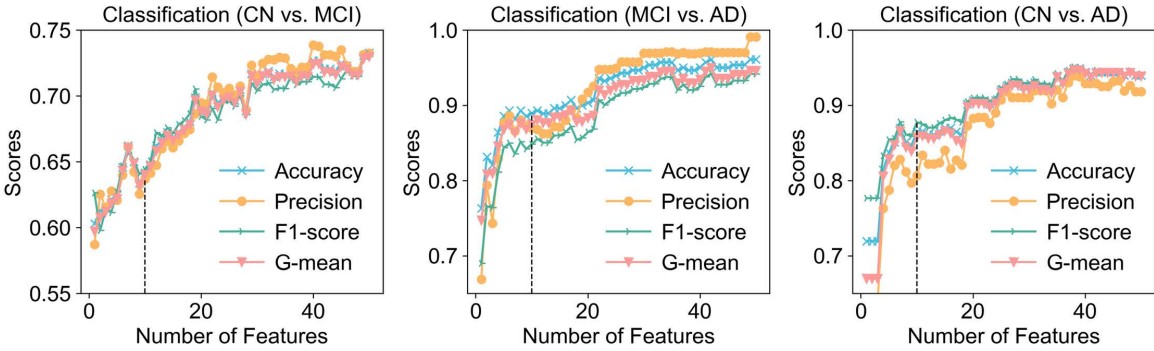

**Fig 8. Classification performance variation with incremental methylation probes (top 50) across three comparative groups.**

The results indicated that model-based integration performed slightly better than concatenation, whereas transformation performed the worst. Across all methods, adding genetic (PRS) and epigenetic (methylation) data to the clinical data (BDI) progressively improved the classification performance (**Fig 10**, S8 Table in S1 File). Specifically, the model-based "C+P+M" strategy significantly outperformed the "C" strategy, increasing MCC, accuracy, and precision by at least 20%, 11%, and 15%, respectively. This demonstrates that multi-omics integration can significantly enhance the accuracy of AD

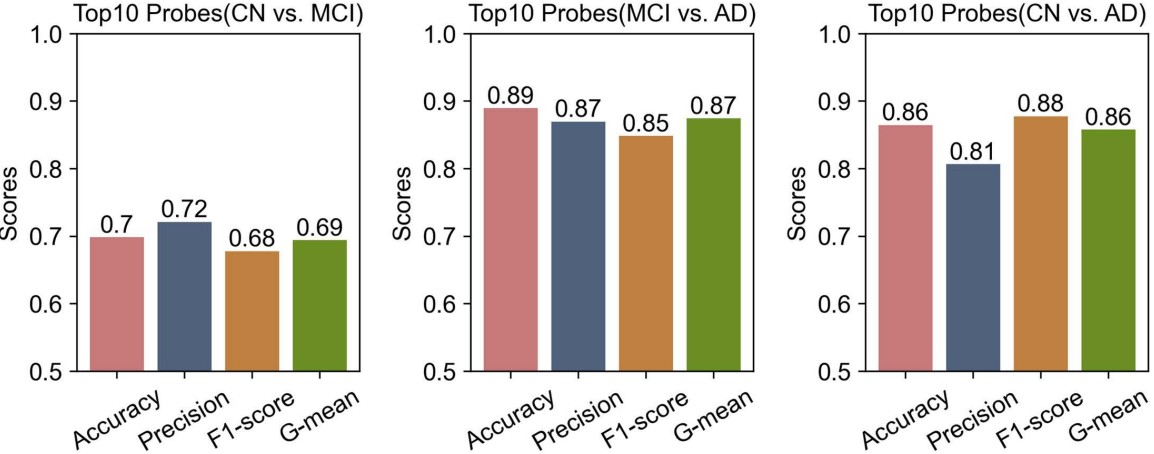

**Fig 9. Classification performance of top 10 methylation probe subsets in three comparative groups.**

**Fig 10. Classification results of multi-omics data integration using conventional machine learning methods.**

prediction. Consistent with previous analyses, all models, regardless of integration strategy, best discriminated CN vs. AD, followed by MCI vs. AD, and CN vs. MCI.

However, traditional ML-based integration approaches often struggle to capture complex interrelationships within multi-omics datasets. These methods typically abstract data into simple matrices or rely on optimization techniques that may not fully account for intricate relationships within multi-omics data. To address this limitation, we developed AD-GCN, a novel graph convolutional neural network designed to explicitly model the topological relationships among multi-omics features. The AD-GCN method integrates three types of omics data — BDI (age, gender, education level, and marital status), LDpred PRS (S2–S4 Tables in S1 File), and the top 10 methylation probe set (S7 Table in S1 File) — by concate-nating them to form graph nodes. The edge set is constructed using sample similarity and KNN distance matrices, with the similarity of 12 cognitive scores between samples serving as edge weights.

Compared with the traditional ensemble methods, AD-GCN achieved superior performance (Fig 11). Specifically, in the CN vs. MCI group, AD-GCN increased the MCC from 0.58 (model-based ensemble) to 0.76, while the other two groups saw MCCs increase to approximately 0.90. This MCC improvement demonstrates the enhanced ability of the AD-GCN to accurately classify both majority and minority classes, indicating a more balanced and robust performance than that of traditional methods. Compared with model-based integration methods (taking model-based integration as an example), AD-GCN demonstrates a reasonable performance trade-off: its runtime is approximately 2.9 times longer than conven-tional approaches (AD-GCN: 13.2s in CN vs. MCI, 11.2s in MCI vs. AD, and 13.5s in CN vs. AD; model-based integration: 5.49s in CN vs. MCI, 3.99s in MCI vs. AD, and 3.9s in CN vs. AD), while peak memory is 1.7 times higher (AD-GCN: 302.14 MiB in CN vs. MCI, 303.00 MiB in MCI vs. AD, and 290.98 MiB in CN vs. AD; model-based integration: 175.95 MiB in CN vs. MCI, 179.60 MiB in MCI vs. AD, and 168.36 MiB in CN vs. AD) (S9 Table in S1 File). This modest elevation in computational resource requirements is justifiable given the significant improvement in classification accuracy (e.g., +9% in CN vs. MCI, +4% in MCI vs. AD, and +4% in CN vs. AD), particularly when considering the inherent computational com-plexity of graph neural networks in processing intricate topological relationships.

## 5. Comparison of AD-GCN ablation study results

Our AD-GCN model incorporates two key improvements: (1) expanding the adjacency matrix using KNN to connect neigh-boring nodes and (2) incorporating 12 cognitive scores as prior knowledge to weight the edges. To evaluate the contribution of these improvements to AD diagnosis, we conducted ablation experiments based on the ADNI data and assessed the

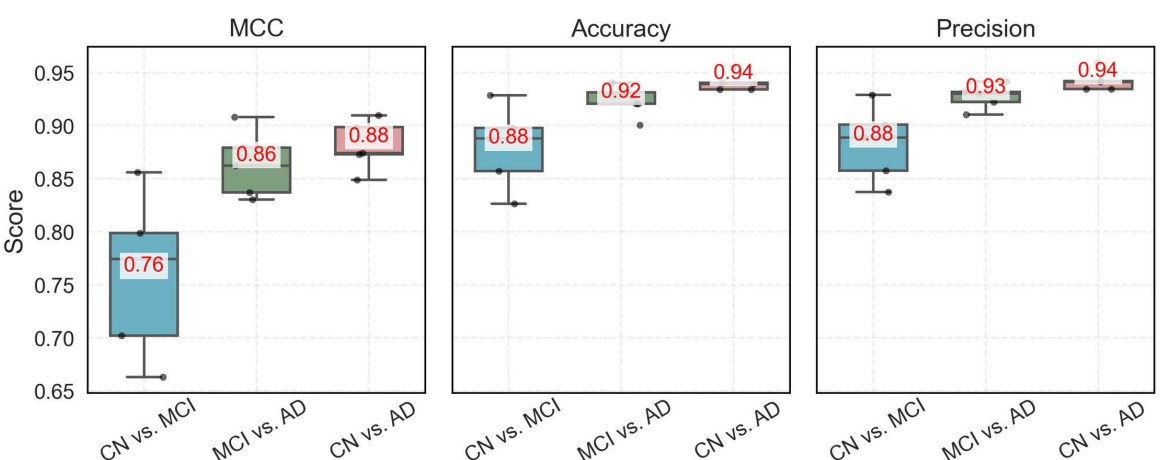

**Fig 11. Classification results of multi-omics data integration using AD-GCN model.**

independent effects by sequentially removing each improvement. Three strategies were compared: (1) a baseline model without improvement, (2) a model with only the KNN improvement, and (3) the full AD-GCN model with both improvements.

As shown in S8 Fig in S1 File, the full AD-GCN achieves the best performance. For example, in the CN vs. MCI group, MCCs were 0.20, 0.72, and 0.76 for strategies 1, 2, and 3, respectively. Similarly, in the MCI vs. AD group, MCCs were 0.54, 0.86, and 0.90, and in the CN vs. AD group, they were 0.70, 0.86, and 0.88. The addition of neighbor nodes substantially improved performance (e.g., a 0.52 increase in MCC for CN vs. MCI), demonstrating its importance. Including cognitive scores provided a further, albeit smaller, improvement (e.g., a 0.04 increase in MCC for CN vs. MCI), confirming their value. These results validate the effectiveness of the proposed approach.

## Discussion

This study integrated multi-omics data, including clinical assessments, genetic variation, and methylation profiles, to enhance AD prediction and address the limitations of traditional clinical assessments for capturing the heterogeneous nature of AD. Our results demonstrate that combining LDpred-derived PRS with clinical cognitive scores significantly improves diagnostic accuracy. Meanwhile, we identified distinct epigenomic signatures associated with AD progression, highlighting the valuable complementary information provided by methylation patterns. To effectively integrate these diverse data, we developed AD-GCN model, a novel graph convolutional neural network that outperformed traditional machine learning ensemble methods, demonstrating the power of deep learning for capturing intricate relationships within multi-omics data and improving AD prediction.

Because of the high-dimensional, low-sample-size (HDLSS) nature of omics data [38], this study employed feature selection and dimensionality reduction techniques. Genomic data were summarized using LDpred PRSs, and the top ten methylation features were selected using filter and embedded methods. We observed progressively improved classification performance in ensemble models with increasing data modalities (C, C+P, and C+P+M), highlighting the contribution of both genetic and epigenetic data to AD diagnosis.

Analysis of the top 20 SNPs contributing most significantly to LDpred PRS in the three comparison groups (CN vs. MCI, MCI vs. AD, and CN vs. AD) identified several key genes, including *APOE*, *APOC1*, *CR1*, *CR1L*, and *TOMM40* (odds ratio > 1.5). These genes play crucial roles in various biological pathways involved in AD pathogenesis. Specifically, *APOE* and *APOC1* are involved in lipid metabolism and influence the accumulation of amyloid-beta (Aβ), a hallmark of AD pathology [39,40]. *CR1* and *CR1L* contribute to immune response and Aβ clearance in the brain. Variations in these genes can affect Aβ removal, thereby modulating AD risk [41]. *TOMM40* plays a role in mitochondrial function and oxidative stress, and its variants have been linked to differences in AD onset and progression, potentially through interactions with *APOE* and its influence on amyloid precursor protein processing [41]. These findings underscore the complex genetic architecture of AD and highlight the involvement of multiple interconnected biological processes.

Similarly, we mapped the three sets of DNA methylation probes (S7 Table in S1 File) to their corresponding genes using genomic coordinates obtained from UCSC (https://genome.ucsc.edu/), including *AACS*, *VTRNA2−1*, and *CYP26C1*, all of which have potential implications in AD. *AACS*, linked to increased AD prevalence, particularly in specific populations, has been shown to be altered in AD case-control brain samples [42]. *VTRNA2−1*, which is not directly linked to AD, plays a role in PKR inhibition, influencing cellular stress responses, and its methylation variation can affect gene expression patterns relevant to neurodegenerative processes [43]. *CYP26C1*, which is involved in retinoic acid metabolism that is crucial for neuronal development, may contribute to AD risk, as abnormal retinoid metabolism has been implicated in neuropsychiatric disorders [44].

To further explore complex multi-omics interactions, we developed the AD-GCN model, which integrates clinical, genetic, and methylation data within a relational graph framework. This graph incorporates sample similarities and k-nearest neighbors using cognitive score correlations as edge weights to form the adjacency matrix of the model. The model leverages both the GCN and GraphSAGE modules for information extraction. The results showed that the AD-GCN model

outperformed ML ensemble methods in using multi-omics data for classification, suggesting that considering the interconnections between omics, rather than merely transforming feature matrices or improving models, provides stronger classification performance. Ablation experiments validated the design choices and robustness of the AD-GCN.

The classification performance consistently followed the following trends: (1) model-based > concatenation-based > transformation-based integration; (2) C + P + M > C + P > C; (3) AD-GCN > ML ensemble; and (4) CN vs. AD > MCI vs. AD > CN vs. MCI. This reinforces the benefits of model-based integration, multi-omics data, and the AD-GCN approach and highlights the increasing diagnostic challenge as the disease progresses from AD to MCI to CN, aligning with clinical observations.

The AD-GCN framework demonstrates superior diagnostic accuracy compared to conventional machine learning integration methods by effectively capturing complex relationships among three readily accessible and cost-effective omics data types: clinical parameters, genetic variants, and methylation profiles. Specifically, our approach utilizes: (1) routine clinical assessments obtainable, (2) a focused panel of 1,322 high-risked SNPs, and (3) a minimal set of 10 methylation probes. This optimized panel design offers significant advantages over WGS or EPIC microarray approaches, achieving comparable diagnostic efficacy while substantially reducing both technical complexity and implementation costs. In clinical application, AD-GCN can achieve patient diagnosis through standardized data acquisition process: Firstly, the clinical data were obtained based on the structured doctor interview scale and the electronic medical record system. Secondly, the peripheral blood of the patient was collected, and then the gene variants and methylation markers were detected by the customized detection panel.

As an innovative multi-omics integration framework, AD-GCN has potential application value for polygenic neurological disorders (NDs), taking Parkinson's disease (PD) as an example: The graph network structure of AD-GCN can naturally accommodate PD-related multi-omics data (such as motor symptom scores, GWAS significant loci, and blood methylation marks) to achieve accurate diagnosis of PD. Furthermore, the analytical framework of AD-GCN offers potential for adaptation to investigate other complex NDs. Such applications could facilitate the identification of disease-specific multi-omics risk factors, which in turn could inform the development of more targeted, accurate, and potentially cost-effective panels for early risk screening ultimately advancing precision diagnosis and management for these complex disorders.

This study has several limitations that warrant further investigation: Firstly, the study cohort predominantly comprised individuals of European ancestry. This limits the generalizability of our findings, as genetic background, environmental exposures, and lifestyle factors influencing AD etiology and biomarker expression can differ substantially across ethnic groups. Consequently, the clinical applicability of the current model requires validation and potentially optimization for non-European populations. Future investigations should prioritize the inclusion of diverse cohorts representing multi-ethnic population. Such expansion will facilitate comparative analyses of genetic background across diverse ancestries, potentially revealing ancestry-specific disease mechanisms. Secondly, although integrating three types of data, our analysis did not incorporate other potentially informative data such as neuroimaging or proteomics. Integrating more data types could enhance model performance but introduces challenges related to increased data dimensionality, necessitating robust feature selection and dimensionality reduction techniques. Application of the AD-GCN framework to integrate multi-omics datasets holds promise for elucidating a more comprehensive understanding of AD pathophysiology. Finally, the cross-sectional design of this study precludes the analysis of temporal dynamics inherent in AD progression. However, establishing longitudinal AD studies cohort and employing temporal modelling approaches (e.g., LSTMs, Transformers, or spatiotemporal GNNs) are crucial to characterize dynamic changes in multi-omics profiles and phenotypic traits over time. Such analyses could significantly enhance the precision of disease trajectory prediction and facilitate the development of timely, personalized interventions.

## Code availability

We provided the source code, which is available at https://github.com/YananLAB/AD-GCN.

## Supporting information

**S1 File.** **S1 Fig.** Heat map of LD values (at chromosome 19) and projection on 2D coordinates for genotype data at overlapping loci between the 1KGP European and ADNI White populations. **S2-S4 Figs.** Linear fits based on LDpred PRS for 12 cognitive scores on CN vs. MCI , MCI vs. AD, and CN vs. AD. **S5-S7 Figs.** The distribution of Fisher scores in CN vs. MCI, MCI vs. AD, and CN vs. AD group. **S8 Fig.** Comparison of AD-GCN ablation study results. Ablation experiments under three strategies all adopt a five-fold cross-validation approach. **S1 Table.** Details of clinical cognitive assessment of ADNI data. **S2-S4 Tables.** LDpred PRS results in CN vs. MCI, MCI vs. AD, and CN vs. AD group. **S5 Table.** Number of probes in information entropy screening process. **S6 Table.** Numbers of methylation probes selected after quality control. **S7 Table.** Top 10 methylation features selected by filtering and embedding methods. **S8 Table.** Evaluation results of different ML integrative methods. **S9 Table.** Comparison of computing requirements of ML integrative methods and AD-GCN model diagnostics. (DOCX)

## Author contributions

**Conceptualization:** Qingyun Li.

**Data curation:** Wei Luo, Haiyan Guo, Chunyan Zhao, Canzhen Yang.

**Formal analysis:** Wei Luo, Yangfan Guo.

**Funding acquisition:** Kai Hu, Yangfan Guo.

**Investigation:** Zilu Li, Qingyun Li.

**Methodology:** Zilu Li.

**Project administration:** Yangfan Guo.

**Software:** Yangfan Guo.

**Supervision:** Xiaoqing Li, Anke Xie, Kai Hu, Yangfan Guo.

**Validation:** Zilu Li.

**Visualization:** Zilu Li.

**Writing – original draft:** Zilu Li.

**Writing – review & editing:** Zilu Li, Kai Hu, Yangfan Guo.

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
