## [Decision Letter · Decision Letter 0]

10 Mar 2025

Dear Dr. Guo,

Thank you for submitting your manuscript to PLOS ONE. After careful consideration, we feel that it has merit but does not fully meet PLOS ONE’s publication criteria as it currently stands. Therefore, we invite you to submit a revised version of the manuscript that addresses the points raised during the review process.

We look forward to receiving your revised manuscript.

Kind regards,

Tao Huang

Academic Editor

PLOS ONE

Journal Requirements:

       1. The National Key Research and Development Program of China (2022YFC2603305)

       2. Key Research and Development Program of Yunnan (202103AQ100002)

       3. Central Funds Guiding the Local Science and Technology Development (202207AB110017)

       4. Yunnan Academician and Expert Workstation (202205 AF150023)

       5. Yunnan Fundamental Research Projects (202201AY070001-192)

       6. The Scientific and Technological Innovation Team in Kunming Medical University (CXTD202215)

Reviewers' comments:

Reviewer's Responses to Questions

**Comments to the Author**

1. Is the manuscript technically sound, and do the data support the conclusions?

Reviewer #1: Yes

Reviewer #2: Yes

Reviewer #3: Yes

Reviewer #4: Yes

Reviewer #5: No

Reviewer #6: Yes

2. Has the statistical analysis been performed appropriately and rigorously?

Reviewer #1: Yes

Reviewer #2: Yes

Reviewer #3: Yes

Reviewer #4: Yes

Reviewer #5: No

Reviewer #6: Yes

3. Have the authors made all data underlying the findings in their manuscript fully available?

Reviewer #1: Yes

Reviewer #2: Yes

Reviewer #3: Yes

Reviewer #4: Yes

Reviewer #5: Yes

Reviewer #6: Yes

4. Is the manuscript presented in an intelligible fashion and written in standard English?

Reviewer #1: Yes

Reviewer #2: Yes

Reviewer #3: Yes

Reviewer #4: Yes

Reviewer #5: Yes

Reviewer #6: Yes

Reviewer #1: The english text sounds well in all paper sections, the topic and idea of the study original and absolutely current. Methodology applied is correct as well as statistic analysis.

I only suggest to better summarize and focus the introduction around the study question.

Reviewer #2: AD-GCN: a novel graph convolutional network integrating multi-omics data for enhanced Alzheimer's disease diagnosis, the manuscript is interesting, it is generally well written, with some grammatical errors. The manuscript is generally well-structured, with each section logically progressing from one topic to the next

Abstract

1. Line 37-38: multi-omics risk factors, such as age, sex, education level, and genomic and epigenomic influences. This should be changed, as when I think of multi-omics I think of genomics, transcriptomics etc, and not of age, sex, etc.

Introduction

1. The authors should include a small introduction about what AD is. An epidemiology of AD in China if this information is possible.

2.Line 96: Contributions should be changed to aims.

Methods

1. Line 108. Remove from the ADNI.

2. What are the 12 cognitive function scores? If possible and this information is available the tests used should be added.

3. The ROSMAP dataset was also obtained from ADNI or from somewhere else? as it is not clear too me.

4. Line 126-127: It would be better to represent the following as a table.

5. What tool was used for statistical analysis? R, excel, SPSS?

6. Line 177: What are the tree ML integration strategies that were implemented?

7. Was previously work done in order to decide the five ML methods that would be applied?

Results

1. Add full names of the cognitive tests used under the table.

2. 1KGP cohorts, why was this cohort not mentioned in the methods? unless I have missed it somewhere.

3. A justification for the threshold values used in the QC process (why specific minor allele frequency or Hardy-Weinberg deviations thresholds were selected) would help readers understand the rationale for these decisions.

4. The comparison between the four PRS models (P+T, LDpred, PRS-CS, and lassosum) is clear. However, a bit more detail on the algorithmic differences of each model would benefit readers who may not be familiar with these methods. For instance, what makes LDpred particularly suitable for this kind of analysis compared to PRS-CS or P+T?

5. AUC analysis is very informative, but it would be helpful to state if improvements are statistically significant (or not), and whether the differences between models (e.g., LDpred vs. lassosum) are meaningful in a clinical context. Adding some statistical tests (e.g., pairwise comparison of AUCs) would provide stronger evidence.

6. The results suggest that PRS models are particularly useful in distinguishing CN from AD, but the manuscript could benefit from a deeper exploration of why this is the case.

7. The authors report KS values and p-values for the linear fits based on LDpred PRS, but a more thorough interpretation of the statistical significance in the context of AD stages would be helpful.

8. The description of methylation data processing using ChAMP and feature selection via Fisher Score and entropy filtering is rigorous, but I would encourage the authors to provide more detail on the thresholds used for quartile-based feature selection. How were these thresholds determined, and do the results vary with different thresholds? This will help readers assess the robustness of your feature selection approach.

9. The AD-GCN as a novel approach for multi-omics integration is particularly interesting. However, the technical description of the graph convolutional network (GCN) could benefit from additional explanation. Specifically, how are cognitive scores incorporated into the graph model? Are these scores treated as node features or edge weights? Further clarification on how the cognitive scores are integrated into the graph structure would help readers better understand this innovative aspect of the model.

Discussion

1. The authors should mention the benefits of this approach to AD and how it may be applied to other NDs as well, as this approach is novel and holds great research potential .

2. The authors should mention the limitations of their work.

General comments.

1. Add full name followed by abbreviation. e.g. line 99.

2. Add full name and links of databases. Also add references to the the database if there is.

3. Sample Homogeneity: The use of only Caucasian participants is understandable, but there should be a more thorough discussion of potential limitations this introduces, particularly in terms of generalizability to other ethnic groups. Given that AD and related diseases have genetic heterogeneity across populations.

4. The figures should be made into a higher resolution.

Reviewer #3: This study presents AD-GCN, a novel graph convolutional network integrating multi-omics data (clinical, genomic, and methylation) to enhance Alzheimer’s disease (AD) diagnosis. The authors demonstrate that multi-omics integration outperforms single-omics approaches and that AD-GCN surpasses traditional machine learning (ML) methods. While the work is innovative and addresses a critical need in AD research, several methodological and presentation issues require clarification and improvement.

Strengths:

The idea of using graph convolutional networks to integrate multi-omics data for AD diagnosis is novel and has the potential to capture complex interactions among different data types.

The use of polygenic risk scores and feature selection methods is well justified and shows promising results in improving classification performance.

The experiments are well designed, with comparisons between single-omics and multi-omics models, as well as ablation studies to evaluate the contributions of different components in the AD-GCN model.

The results demonstrate that the AD-GCN model outperforms traditional machine learning ensemble methods, highlighting its potential clinical utility.

Weaknesses:

The study is limited to Caucasian individuals, which may limit the generalizability of the findings to other racial groups. Broader inclusion of diverse populations would strengthen the study.

The availability of public datasets is acknowledged as a limitation. The ADNI cohort includes only 35 AD cases, which raises concerns about statistical power and generalizability. Expanding the number of accessible datasets could further validate and improve the model.

The biological interpretation of the selected methylation markers could be strengthened with additional functional validation studies.

Specific Comments:

1、Abbreviations (e.g., BDI, MCC) should be defined at first mention.

2、The Introduction section provides a comprehensive background on AD and multi-omics data integration. However, more details on existing graph convolutional network approaches for similar applications could be included to position the current work more clearly within the field.

3、LDpred outperforms other PRS methods, but the manuscript lacks a discussion of why LDpred is better suited for AD risk modeling compared to alternatives like PRS-CS.

4、Validate Generalizability: Test AD-GCN on ROSMAP or another independent cohort.

5、The Discussion overemphasizes technical contributions; a deeper exploration of biological insights and future research directions could be further expanded. .

Recommendation:

Overall, this is an interesting and potentially impactful study. With some minor revisions to address the limitations and clarify certain aspects of the methods, I recommend this manuscript for publication in PLOS ONE after minor revisions. The novel approach and promising results indicate the potential for significant advances in AD diagnosis and personalized medicine.

Reviewer #4: The study presents a novel graph convolutional network (GCN) approach for integrating clinical, genetic, and methylation data, offering a promising method for AD diagnosis. However, a few refinements could further strengthen the manuscript:

1. Clarity in Structure – The Results and Methods sections are interwoven, making it challenging to follow the study’s flow. A clearer separation of these sections would improve readability and comprehension.

2. GWAS and PRS Methodology – The description of GWAS and polygenic risk score (PRS) analysis needs more clarity. Instead of performing GWAS on a relatively small dataset, the authors could consider leveraging publicly available GWAS summary statistics to enhance the robustness and interpretability of findings. Similarly, more details on the thresholds and criteria used in PRS calculations and random forest feature selection would improve reproducibility.

3. Clinical Applicability – The Discussion section could be expanded to address how AD-GCN could be integrated into clinical practice and what potential limitations or challenges might arise in real-world applications.

4. Some sentences are not very fluent and the figures are not very clear.

These refinements would help better highlight the study’s contributions and improve its overall impact.

Reviewer #5: The authors developed a machine learning tool named AD-GCN for Alzheimer's disease diagnosis using clinical data, genomic variants, and DNA methylation data. The biggest issue is there are only 35 AD cases that have all three types of omics data, such a small sample size is not enough for the model to extract features. In addition, CN cases were 6 times more than AD cases, and MCI cases were 10 times more than AD cases in the ADNI dataset. The sample size imbalance in these groups will cause super-control bias in the model prediction. The average AUC of all 5 methods in MCI vs. AD and CN vs. AD are lower than 0.75, the AUC of two methods in CN vs. AD even fluctuates around 0.5 (Figure 5). This means that the above models are useless for AD diagnosis. Although the author did many analyses and evaluations of these models, the insufficient sample size is the fatal flaw of the manuscript.

Reviewer #6: The manuscript presents a well-executed study on integrating multi-omics data for Alzheimer’s disease classification using a novel graph convolutional network (AD-GCN). The work is methodologically strong, and the results are promising. The comparisons with traditional machine learning models highlight the advantages of AD-GCN in handling complex multi-omics relationships.

Strengths:

• The use of a graph convolutional network (GCN) to integrate omics data is an interesting and innovative approach.

• The comparison between single-omics and multi-omics classification methods is well executed and demonstrates the benefits of data integration.

• The inclusion of polygenic risk scores and methylation features strengthens the study’s ability to capture genetic and epigenetic risk factors for AD.

Areas for Improvement:

• Biological Interpretation: The study identifies key genetic (APOE, CR1, TOMM40) and epigenetic (AACS, VTRNA2-1, CYP26C1) markers, but their roles in AD pathogenesis are not discussed in depth. A more detailed interpretation would be valuable.

• Justification for Adjacency Matrix Construction: The authors use sample similarity and cognitive scores to build the adjacency matrix in AD-GCN. Could they clarify why this approach was chosen over other possible graph structures (e.g., biological pathway-driven networks)?

• Computational Efficiency: AD-GCN is shown to outperform traditional ML models in classification accuracy, but the manuscript does not discuss its computational requirements. Providing details on runtime, memory usage, and scalability would be helpful for researchers looking to apply this method.

• Generalizability and Population Diversity: The study is based on ADNI and ROSMAP, which primarily include Caucasian participants. Have the authors considered validating the model on more diverse datasets?

• Reproducibility: The authors mention that their code is available on GitHub. Could they confirm whether preprocessing scripts and detailed environment configurations are included to allow full replication of results?

Final Recommendation:

This manuscript presents a valuable contribution to computational approaches for AD diagnosis and is well suited for publication after minor revisions. Addressing the interpretability of genetic findings, discussing computational feasibility, and clarifying the adjacency matrix construction would further strengthen the study.

**Do you want your identity to be public for this peer review?** For information about this choice, including consent withdrawal, please see our Privacy Policy

Reviewer #1: No

Reviewer #2: No

Reviewer #3: No

Reviewer #4: No

Reviewer #5: No

Reviewer #6: **Yes: ** Ece Eldem

---

## [Author Response · Author response to Decision Letter 1]

17 Apr 2025

Dear Editors and Reviewers:

We appreciate you and the reviewers for your precious time in reviewing our paper and providing valuable comments. It was your valuable and insightful comments that led to possible improvements in the current version. I would like to express my sincere gratitude for your time and effort in handling my manuscript, “AD-GCN: A novel graph convolutional network integrating multi-omics data for enhanced Alzheimer’s disease diagnosis”, and for providing me with the opportunity to revise it. I have completed the major revision of the manuscript in strict accordance with the reviewers’ requirements.

For each reviewer’s comments, I have provided detailed and respectful responses. I have carefully addressed each and every comment in a point- by-point manner and made all the necessary adjustments to the manuscript.

Moreover, I stated that the funders had no role: The funders had no role in study design, data collection and analysis, decision to publish, or preparation of the manuscript. I have added a figure label and title for Figures 1 to 11 in my main manuscript. And I have added Supporting Information captions at the end of my manuscript, and updated all in-text citations to match accordingly.

Reviewer #1

Comment 1: The english text sounds well in all paper sections, the topic and idea of the study original and absolutely current. Methodology applied is correct as well as statistic analysis. I only suggest to better summarize and focus the introduction around the study question.

Author Response 1: We sincerely appreciate your positive assessment of our manuscript’s originality, methodology, and statistical rigor. In response to your valuable suggestion to better summarize the Introduction Section, we have implemented the following targeted revisions: (1) included a small introduction about what AD is and an epidemiology of AD in China (Lines 58-62 in Manuscript); (2) deleted sentences with little relevance (“ABCA7 and CLU. The APOE ɛ4 allele is...”); (3) the methodological background has been revised to clarify the positioning of current work within existing machine learning ensemble and GCN-based multi-omics integration approaches (Lines 83-104 in Manuscript).

Reviewer #2

Comment 1 [Abstract-1]: Line 37-38: multi-omics risk factors, such as age, sex, education level, and genomic and epigenomic influences. This should be changed, as when I think of multi-omics I think of genomics, transcriptomics etc, and not of age, sex, etc.

Author Response 1: We agree that “multi-omics” should refer specifically to biological data types such as genomics, transcriptomics, and epigenomics, while demographic factors like age, sex, and education level should be categorized separately. We have revised the sentence accordingly to clarify this distinction (Lines 36-38 in Manuscript):

“Alzheimer’s disease (AD) etiology is complex, influenced by demographic risk factors such as age, sex, and educational level, alongside multi-omics factors derived from genomics, transcriptomics, and epigenomics.”

Comment 2 [Introduction-1]: The authors should include a small introduction about what AD is. An epidemiology of AD in China if this information is possible.

Author Response 2: We have added a concise introduction to Alzheimer’s disease, including its definition and epidemiology in China, to provide context for the study [1]. This addition highlights the importance of our research in addressing this public health challenge (Lines 58-62 in Manuscript).

Reference:

[1] Longfei J, Yifeng D, Lan C, et al. Prevalence, risk factors, and management of

dementia and mild cognitive impairment in adults aged 60 years or older in China: a crosssectional study[J]. The Lancet Public Health, 2020, 5(12):661-671.

Comment 3 [Introduction-2]: Line 96: Contributions should be changed to aims.

Author Response 3: We have revised the “contributions” section to “aims” to better align with the following content.

Comment 4 [Methods-1]: Line 108. Remove from the ADNI.

Author Response 4: We have removed the phrase “from the ADNI” from Line 108 to improve clarity and conciseness.

Comment 5 [Methods-2]: What are the 12 cognitive function scores? If possible and this information is available the tests used should be added.

Author Response 5: We have revised the sentence to clarify that the 12 cognitive function scores are derived from specific tests (e.g., CDRSB, ADAS11, MMSE, RAVLT, etc.) and indicated that detailed descriptions of these tests are provided in S1 Table in S1 File (Lines 132-134 in Manuscript).

Comment 6 [Methods-3]: The ROSMAP dataset was also obtained from ADNI or from somewhere else? as it is not clear too me.

Author Response 6: To clarify, the ADNI dataset was obtained from the Alzheimer’s Disease Neuroimaging Initiative (http://adni.loni.usc.edu), and the ROSMAP dataset was obtained from the Synapse (https://www.synapse.org/Synapse:syn3219045) with appropriate data access permissions. The ROSMAP data was used in GWAS analyses to provide effect sizes for SNPs under different disease stages of the ADNI cohort. I have added this clarification to the Materials and Methods section (Lines 138-142 in Manuscript).

Comment 7 [Methods-4]: Line 126-127: It would be better to represent the following as a table.

Author Response 7: We agree that this would enhance the clarity and readability of the information. We have revised the manuscript by replacing the text with a table titled “Table 1. Case Distribution in ADNI and ROSMAP Cohorts.” (Line 159 in Manuscript).

Comment 8 [Methods-5]: What tool was used for statistical analysis? R, excel, SPSS?

Author Response 8: To clarify, all statistical analyses, including the application of the KNN algorithm for filling missing values and the Kruskal-Wallis test and chi-square test for group comparisons, were performed using the R software (version 4.3) (Lines 156-158 in Manuscript).

Comment 9 [Methods-6]: Line 177: What are the three ML integration strategies that were implemented?

Author Response 9: We have revised the text to explicitly list the three integration methods (concatenation-based integration, transformation-based integration, and model-based integration) at the beginning of the section for improved readability (Lines 222-224 in Manuscript).

Comment 10 [Methods-7]: Was previously work done in order to decide the five ML methods that would be applied?

Author Response 10: The choice of the five ML methods (KNN, LR, Bayes, SVM, and RF) was based on their fundamental nature and widespread use in the field of machine learning [1]. They represent diverse learning paradigms, including similarity-based (KNN), error-based (LR), probability-based (Bayes), kernel-based (SVM), and information-based (RF) approaches, making them suitable for evaluating the effectiveness of our model-based integration strategy.

Additionally, while advanced methods like XGBoost and AdaBoost are powerful, they often require extensive hyperparameter tuning and computational resources. Given our focus on evaluating fundamental integration strategies, we prioritized simpler yet representative methods to ensure a clear and interpretable comparison.

To address this point more explicitly in the manuscript, we have revised the relevant sentence in the Methods section as follows (Lines 229-232 in Manuscript):

“Finally, model-based integration uses separate classifiers for each omics data with five widely-used ML methods (KNN, LR, Bayes, SVM, and RF), selected for their diverse learning paradigms and fundamental nature in machine learning.”

Reference:

[1] Reel PS, Reel S, Pearson E, et al. Using machine learning approaches for multi-omics data analysis: A review. Biotechnology Advances. 2021;49:107739. doi:10.1016/j.biotechadv.2021.107739

Comment 11 [Results-1]: Add full names of the cognitive tests used under the table.

Author Response 11: We have updated the manuscript to include the full names of the cognitive tests used under the table for clarity (Lines 277-284 in Manuscript).

Comment 12 [Results-2]: 1KGP cohorts, why was this cohort not mentioned in the methods? unless I have missed it somewhere.

Author Response 12: Thank you for your suggestion. We have supplemented the introduction of 1KGP data and the role of the LD panel in the PRS framework in the Materials and methods section (Lines 146-150 and 185-189 in Manuscript).

Comment 13 [Results-3]: A justification for the threshold values used in the QC process (why specific minor allele frequency or Hardy-Weinberg deviations thresholds were selected) would help readers understand the rationale for these decisions.

Author Response 13: We have revised the manuscript to include the rationale behind the selection of these thresholds [1-4]. This addition aims to provide readers with a clearer understanding of the scientific basis for our QC decisions (Lines 165-174 in Manuscript).

Reference:

[1] Marees AT, De KH, Stringer S, et al. A tutorial on conducting genome‐wide association studies: Quality control and statistical analysis. International journal of methods in psychiatric research. 2018;27(2):e1608. doi:10.1002/mpr.1608

[2] Chang CC, Chow CC, Tellier LC, et al. Second-generation PLINK: rising to the challenge of larger and richer datasets. Gigascience. 2015 Dec;4(1):s13742-015. doi:10.1186/s13742-015-0047-8

[3] Wigginton JE, Cutler DJ, Abecasis GR. A note on exact tests of Hardy-Weinberg equilibrium. The American Journal of Human Genetics. 2005 May 1;76(5):887-93. doi:10.1086/429864

[4] Loh PR, Tucker G, Bulik-Sullivan BK, et al. Efficient Bayesian mixed-model analysis increases association power in large cohorts. Nature genetics. 2015 Mar;47(3):284-90. doi:10.1038/ng.3190

Comment 14 [Results-4]: The comparison between the four PRS models (P+T, LDpred, PRS-CS, and lassosum) is clear. However, a bit more detail on the algorithmic differences of each model would benefit readers who may not be familiar with these methods. For instance, what makes LDpred particularly suitable for this kind of analysis compared to PRS-CS or P+T?

Author Response 14: We fully concur with this perspective and have supplemented in the manuscript the rationale for why LDpred is particularly suitable for this kind of analysis (Lines 364-372 in Manuscript):

“From the perspective of shrinkage mechanisms, P+T retains significant SNPs through hard thresholding but loses weak-effect signals, while PRS-CS employs global continuous shrinkage that excessively compresses moderate-effect variants. Lasso-type methods like lassosum analyze only partial SNPs in high-LD regions through regularization. In contrast, LDpred achieves localized adaptive shrinkage by explicitly modeling LD structures, which not only coordinates the adjustment of SNP effect sizes within high-LD regions but also preserves subthreshold genetic signals, thereby demonstrating higher variance explained and classification accuracy in AD risk prediction.”

Comment 15 [Results-5]: AUC analysis is very informative, but it would be helpful to state if improvements are statistically significant (or not), and whether the differences between models (e.g., LDpred vs. lassosum) are meaningful in a clinical context. Adding some statistical tests (e.g., pairwise comparison of AUCs) would provide stronger evidence.

Author Response 15: We sincerely appreciate your suggestion to evaluate the statistical significance of AUC differences between models. However, after careful consideration, we decided not to perform formal hypothesis testing for the following reasons:

(1)Insufficient statistical power. Nonparametric tests (e.g., the Wilcoxon test) have very low sensitivity at n=5 and are difficult to detect true differences (prone to false negatives). Parametric tests (e.g., the t-parameter) require that the data be approximately normally distributed, but n=5 is not a reliable test for normality.

(2)The p-value is difficult to interpret. With small samples, p-values are very sensitive to extreme values, which can result in significant results by chance (false positives) or mask true differences (false negatives).

(3)Additionally, the most works of polygenic risk score performance evaluations exclusively utilize AUC metrics without incorporating p-values in their assessments [1-3].

So we choose to list the mean AUC of each group to compare the differences of the four methods, as shown in Table 3 (Lines 353-354 in Manuscript).

Reference:

[1]Ni G, Zeng J, Revez JA, et al. A comparison of ten polygenic score methods for psychiatric disorders applied across multiple cohorts. Biological psychiatry. 2021 Nov 1;90(9):611-20.

[2]Thompson DJ, Wells D, Selzam S, et al. A systematic evaluation of the performance and properties of the UK Biobank Polygenic Risk Score (PRS) Release. Plos one. 2024 Sep 18;19(9):e0307270.

[3]Lennon NJ, Kottyan LC, Kachulis C, et al. Selection, optimization and validation of ten chronic disease polygenic risk scores for clinical implementation in diverse US populations. Nature medicine. 2024 Feb;30(2):480-7.

Comment 16 [Results-6]: The results suggest that PRS models are particularly useful in distinguishing CN from AD, but the manuscript could benefit from a deeper exploration of why this is the case.

Author Response 16: We sincerely appreciate your insightful comment regarding the need to further explore the underlying mechanisms behind PRS’s discriminative performance between CN and AD. In response to your suggestion, we have expanded our discussion in the Results to provide a more comprehensive explanation. The additional explanation in the original article is as follows (Lines 357-362 in Manuscript):

“This excellent ability of PRS in distinguishing CN from AD stems from the polygenic genetic nature of AD. As a highly inherited disease, AD is driven by both high-effect variants such as APOE-ε4 and numerous small-effect loci. PRS integrates these genetic effects to give AD patients a significantly higher cumulative risk than the CN population with a protective or neutral genetic background.”

Comment 17 [Results-7]: The authors report KS values and p-values for the linear fits based on LDpred PRS, but a more thorough interpretation of the statistical significance in the context of AD stages would be helpful.

Author Response 17: In response, we have conducted a more thorough interpretation of the statistical significance within the context of AD progression stages. Specifically, we have: (1) elaborated on the respective clinical implications of both KS values and p-values, (2) provided detailed exemplification using ADAS13 as a representative cognitive metric, and (3) ultimately demonstrated the robust correlation between LDpred-derived PRS and cognitive function phenotypes (Lines 376-393 in Manuscript).

Comment 18 [Results-8]: The description of methylation data processing using ChAMP and feature selection via Fisher Score and entropy filtering is rigorous, but I would encourage the authors to provide more detail on the thresholds used for quartile-based feature selection. How were these thresholds determined, and do the results vary with different thresholds? This will help readers assess the robustness of your feature selection approach.

Author Response 18: We appreciate the constructive suggestions on our feature selection methodology. In response, we have provided a detailed rationale for selecting quartile-based thresholds in both Fisher Score and entropy filtering approaches (Lines 404-414 in Manuscript).

The approximately normal distribution of Fisher Scores (S5-S7 Figs in S1 File) makes quartile thresholds statistically appropriate. In the Fisher Score-filtered probe set, we further applied entropy-based filtering to remove probes with zero information gain. A comparison showed that, across the three groups, the number of probes with information gain greater than zero was similar to the number of probes within the top 25%–100% quartile range (S2 Table in S1 File). Given the minimal differences between the two methods, we adopted the 25% threshold for both to streamline the process and maintain consistency.

Comment 19 [Results-9]: The AD-GCN as a novel approach for multi-omics integration is particularly interesting. However, the technical description of the graph convolutional network (G

---

## [Decision Letter · Decision Letter 1]

7 May 2025

AD-GCN: A novel graph convolutional network integrating multi-omics data for enhanced Alzheimer's disease diagnosis

PONE-D-25-05441R1

Dear Dr. Guo,

We’re pleased to inform you that your manuscript has been judged scientifically suitable for publication and will be formally accepted for publication once it meets all outstanding technical requirements.

Kind regards,

Tao Huang

Academic Editor

PLOS ONE

Additional Editor Comments (optional):

Reviewers' comments:

Reviewer's Responses to Questions

**Comments to the Author**

Reviewer #2: (No Response)

Reviewer #3: All comments have been addressed

Reviewer #6: All comments have been addressed

2. Is the manuscript technically sound, and do the data support the conclusions?

Reviewer #2: Yes

Reviewer #3: Yes

Reviewer #6: Yes

3. Has the statistical analysis been performed appropriately and rigorously?

Reviewer #2: Yes

Reviewer #3: Yes

Reviewer #6: Yes

4. Have the authors made all data underlying the findings in their manuscript fully available?

Reviewer #2: Yes

Reviewer #3: Yes

Reviewer #6: Yes

5. Is the manuscript presented in an intelligible fashion and written in standard English?

Reviewer #2: Yes

Reviewer #3: Yes

Reviewer #6: Yes

Reviewer #2: I thank the authors for the changes made to the manuscript. I have no further comments or concerns to add and I am happy with the revisions made.

Reviewer #3: (No Response)

Reviewer #6: I appreciate the authors’ thorough and thoughtful revisions. The updated manuscript has addressed the concerns raised in the previous round. The manuscript now reads more clearly, and the methodological explanations are better justified. I find the revised version technically sound and suitable for publication.

**Do you want your identity to be public for this peer review?** For information about this choice, including consent withdrawal, please see our Privacy Policy

Reviewer #2: No

Reviewer #3: No

Reviewer #6: No

---

## [Editor Report · Acceptance letter]

PONE-D-25-05441R1

PLOS ONE

Dear Dr. Guo,

I'm pleased to inform you that your manuscript has been deemed suitable for publication in PLOS ONE. Congratulations! Your manuscript is now being handed over to our production team.

Kind regards,

on behalf of

Dr. Tao Huang

Academic Editor

PLOS ONE